# Context-Adaptive Multi-Prompt Embedding with Large Language Models for Vision-Language Alignment

**Dahun Kim**

Google DeepMind

**Anelia Angelova**

Google DeepMind

## Abstract

We propose Context-Adaptive Multi-Prompt Embedding, a novel approach to enrich semantic representations in vision-language contrastive learning. Unlike standard CLIP-style models that rely on a single text embedding, our method introduces multiple structured prompts, each containing a distinct adaptive token that captures diverse semantic aspects of the input text. We leverage a pretrained LLM as the text encoder within the CLIP framework, processing all prompts jointly in a single forward pass. The resulting prompt embeddings are combined into a unified text representation, enabling semantically richer alignment with visual features. To further promote semantic diversity and representation quality, we incorporate a diversity regularization loss and a negation-aware loss, encouraging specialization across prompts and improving contrastive discrimination. Our method achieves consistent improvements on both image-text and video-text retrieval benchmarks.

## 1 Introduction

Contrastive vision-language models, such as CLIP (Radford et al., 2021), have become foundational for zero-shot image-text and video-text retrieval. These models align visual and textual representations by maximizing similarity between paired modalities. However, most approaches rely on a single text embedding per input, which can limit the ability to capture the full range of semantic cues in natural language descriptions.

Effective alignment between text and image or video often requires nuanced, multi-aspect matching. A caption may describe the main subject, relevant objects, or background context, highlighting the need for expressive and semantically rich text embeddings. Pretrained Large Language Models (LLMs), especially decoder-only architectures like GPT (Achiam et al., 2023; Dubey et al., 2024; Team et al., 2024), offer strong semantic reasoning capabilities, making them promising candidates for text encoders in this setting.

However, using LLMs for CLIP requires careful adaptation. The causal structure of decoder-only LLMs limits their ability to summarize entire sequences using standard pooling strategies like first-token or mean pooling. To overcome this, recent methods have proposed prompt-based last-token pooling (Jiang et al., 2023a; Lei et al., 2024a), which guide the model to produce more meaningful representations by prompting it to summarize the input at the final token position. While these methods improve representation quality, they typically use a single prompt or fixed prompt templates, limiting adaptability. This restricts their ability to capture diverse semantic views aligned with visual inputs. Meanwhile, retrieval tasks often require recognizing multiple aspects of a visual-text pair, such as subjects, objects, or scene-level cues.

To address this, we propose Context-Adaptive Multi-Prompt Embedding, a method that introduces multiple structured prompts, each with a distinct adaptive token trained to specialize in a unique semantic aspect of the input. These prompts are processed by a pretrained LLM, and their embeddings are combined into a unified representation aligned with the visual embedding. To further enrich alignment, we introduce two additional learning objectives: a diversity regularization loss that encourages semantic diversity across

prompt embeddings, and a negation-aware loss that introduces additional contrastive signals using negated prompt variants. Our method does not require additional text-only pretraining (e.g. SimCSE (Gao et al., 2021)) and generalizes well to both image-text and video-text retrieval. Extensive experiments on standard retrieval benchmarks demonstrate consistent performance gains over CLIP-style baselines and ablation variants, confirming the benefit of semantically diverse alignment in vision-language contrastive learning.

## 2 Related Work

### 2.1 Vision-Language Contrastive Learning

Contrastive learning has become a foundational approach for aligning visual and textual modalities, driven by the success of CLIP (Radford et al., 2021) and its variants. These models optimize contrastive losses over large-scale image-text or video-text pairs, enabling robust retrieval performance across domains. Subsequent works have explored architectural and training improvements (Li et al., 2021; Jia et al., 2021; Yu et al., 2022), as well as extensions to the video domain by incorporating temporal components into the dual-encoder architecture (Xu et al., 2021; Luo et al., 2022; Wang et al., 2022; Kim et al., 2025). While these methods are effective, they typically rely on simple text encoders. Our work enhances this framework by leveraging rich language representations from pretrained LLMs through multiple context-adaptive prompt embeddings, optimized within a vision-language contrastive setting.

### 2.2 Decoder-Only LLMs for Text Embeddings

Decoder-only large language models (LLMs) such as GPT (Brown et al., 2020), LLaMA (Touvron et al., 2023), Gemini (Comanici et al., 2025) and Gemma (Team et al., 2024) have demonstrated impressive capabilities in generative tasks. However, leveraging them as effective encoders for text representation remains challenging due to their unidirectional causal attention mechanism. A common approach is to use the hidden state of the final token as a sentence embedding (Neelakantan et al., 2022; Ma et al., 2023; Wang et al., 2023), though this can yield suboptimal semantic summarization. Several studies have proposed architectural adjustments to address this issue, such as relaxing the attention mask or enabling partial bidirectional attention in later layers (Li et al., 2023; Dukić & Šnajder, 2024). Other works explore prompt-based strategies (Jiang et al., 2023b; Lei et al., 2024b; Zhuang et al., 2024; Zhang et al., 2024) to better guide the final token's representation. PromptEOL (Jiang et al., 2023b) proposes summarization prompts such as "This image means just in one word:" to steer the final token representation toward semantic abstraction. MetaEOL (Lei et al., 2024b) expands this idea using a diverse set of fixed task-oriented prompts. Additional strategies include Echo (Springer et al., 2024), which duplicates the input to simulate bidirectional context, and LLM2Vec (BehnamGhader et al., 2024), which enhances representations through hybrid attention during supervised contrastive learning. While these methods are designed for text-only embedding tasks, we extend this direction to vision-language contrastive learning by introducing multiple context-adaptive prompts that are learned to enhance semantic diversity and improve alignment with visual content.

### 2.3 LLMs in Vision-Language Contrastive Models

Recent efforts explore the integration of LLMs as text encoders in vision-language contrastive models. JinaCLIP (Xiao et al., 2024) and LLM2CLIP (Wu et al.) replace the CLIP text encoder with off-the-shelf LLMs such as Jina-v2 or OPT (Zhang et al., 2022). These methods typically apply additional text-text contrastive learning such as SimCSE (Gao et al., 2021) before adapting to vision-language training. E5-V (Jiang et al., 2024b) uses a fixed prompt to extract LLM embeddings and fuses them with frozen vision features at the Multimodal LLM layer. While these works demonstrate that LLMs can be effective text encoders for CLIP-style tasks, most approaches rely on single fixed prompts or require pretraining on additional text-text contrastive tasks. Our approach directly learns multiple context-adaptive prompt embeddings within a vision-language contrastive learning. We further

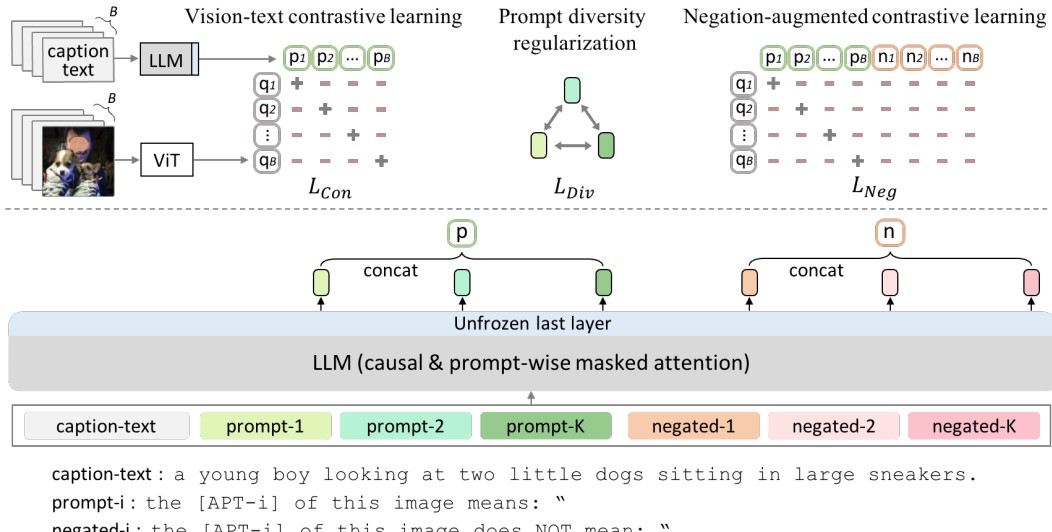

Figure 1: **Overview of our method.** Given an input text, we construct multiple structured prompts, each with a distinct adaptive prompt token [APT-i] (Sec. 3.2). The entire sequence is processed jointly in a single LLM forward pass via prompt-wise attention masking (Sec. 3.3). This yields $K$ embeddings of size $D/K$, concatenated into a $D$-dimensional text embedding. A ViT encoder with attention pooling (Sec. 3.1) produces a matching image or video embedding. Training combines contrastive loss ($L_{con}$) with a prompt diversity regularization loss ($L_{div}$, Sec. 3.4) and a negation-aware loss ($L_{neg}$, Sec. 3.5) to encourage semantic variation and incorporate contrastive signals from negated prompts.

enhance this framework with prompt diversity regularization and negation-aware prompt embeddings, leading to more discriminative representations for vision-language alignment.

## 3 Method

We introduce Context-Adaptive Multi-Prompt Embedding for vision-language alignment, a novel approach designed to enhance semantic richness in vision-language contrastive learning frameworks such as CLIP. Our method leverages pretrained Large Language Models (LLMs) to extract multiple diverse semantic embeddings for each textual input, significantly improving semantic alignment with visual information.

### 3.1 Leveraging Pretrained LLMs for Text Embeddings

Standard CLIP models typically employ Transformer architectures with bidirectional attention (similar to BERT (Devlin et al., 2018)) for text encoding. These models prepend a special token, such as [CLS], to the input text and utilize the hidden state of this token (first-token pooling) as the text embedding. In contrast, recent large language models (LLMs), such as GPT variants, adopt decoder-only architectures with causal attention. Due to the unidirectional nature of causal attention, first-token pooling is ineffective as the embedding of the first token cannot summarize subsequent text.

To obtain high-quality embeddings from decoder-only LLMs, alternative pooling strategies are required. Last-token pooling is intuitively suitable for such models, as only the last position potentially contains information about the entire text. However, decoder-only models trained via next-token prediction inherently align the embedding at the last position primarily toward predicting subsequent tokens, rather than summarizing previous semantic content. To mitigate this, prompt-based last pooling methods, such as PromptEOL (Jiang

et al., 2023a), introduce explicitly structured prompts to guide the model toward semantic summarization at the last position. PromptEOL employs prompts such as: `"[input_text].` `This image means just in one word:"`, thereby instructing the model to succinctly summarize input text semantics. Similarly, MetaEOL (Lei et al., 2024b) introduces multiple manually designed prompts to generate richer semantic embeddings.

Adopting pretrained LLMs as textual encoders in vision-language frameworks such as CLIP provides significant advantages. These LLMs have been extensively pretrained on vast textual datasets, equipping them with rich semantic knowledge and strong generalization capabilities. By leveraging such pretrained LLMs, our model can effectively transfer their deep semantic understanding to vision-text alignment tasks.

## 3.2 Context-Adaptive Multi-Prompt Embedding

Existing prompt-based embedding methods, such as PromptEOL (Jiang et al., 2023a) and MetaEOL (Lei et al., 2024a), rely on fixed, manually designed prompts to guide representation learning. While effective in text-only settings, these approaches limit flexibility when extended to vision-language contrastive learning. In vision-language tasks, textual embeddings are directly trained to align with visual features. This setup presents a new opportunity: prompts can be learned and adapted during training to better reflect the semantic alignment between text and image or video content.

To leverage this, we introduce a context-adaptive multi-prompt embedding strategy that dynamically learns K distinct prompts during vision-text contrastive training. Each prompt contains an Adaptive Prompt Token [APT-i], a special token whose representation is learned to capture distinct semantic aspects of the input text. These tokens are trained in the context of aligning textual and visual representations, enabling the prompts to specialize in diverse and complementary interpretations. The structured prompt follows the format:

`"[input_text]. The [APT-i] of this image means:"`

Each prompt is processed by the pretrained LLM, and we apply last-token pooling to extract a single embedding per prompt. This embedding is then passed through a projection layer to produce an embedding of size D/K, where D is the target embedding size that matches the CLIP visual embedding. The K projected embeddings are concatenated along the channel dimension to form a single text embedding of size D. This concatenation preserves the distinct semantics of each prompt while enabling specialized alignment with the visual representation. Since contrastive learning operates through element-wise dot product between text and visual embeddings, each prompt embedding is matched to a specific channel segment of the visual embedding. This design encourages each [APT-i] to specialize and align with distinct visual-semantic concepts, as demonstrated later in Sec. 4.5.

Both text and vision embeddings are L2 normalized resulting in $p$ and $q$, respectively. The cosine similarity of the embeddings in batch $B$, scaled by a learnable temperature $\tau$ forms the input to the InfoNCE loss (Oord et al., 2018; Radford et al., 2021), defined as:

$$L_{T2I} = -\frac{1}{B} \sum_{i=1}^{B} \log \left( \frac{exp(p_i q_i / \tau)}{\sum_{j=1}^{B} exp(p_i q_j / \tau)} \right), L_{I2T} = -\frac{1}{B} \sum_{i=1}^{B} \log \left( \frac{exp(q_i p_i / \tau)}{\sum_{j=1}^{B} exp(q_i p_j / \tau)} \right). \quad (1)$$

The final contrastive loss is averaged as $L_{con} = (L_{T2I} + L_{I2T})/2$.

## 3.3 Efficient Prompt-Wise Attention Masking

Computing embeddings from multiple distinct prompts conventionally requires multiple forward passes through the LLM, which is computationally inefficient. To address this, we introduce prompt-wise attention masking, enabling efficient computation of all embeddings within a single forward pass. Specifically, we concatenate multiple prompts into one sequence. A representative concatenated prompt is:

`[input_text]. The [APT-1] of this image means:" [APT-2] of this image means:" ...` `[APT-K] of this image means:"`

In this concatenated prompt, the initial prefix tokens ([input_text]. The) are globally accessible and shared among all prompt segments. However, the subsequent prompt-specific segments ([APT-i] of this image means:") are masked from attending to tokens in other segments ([APT-j] of this image means:"). Then, each prompt's embedding is independently computed from the last token (") position of each segment.

To further optimize training efficiency and model size, we freeze most layers of the pretrained LLM, only unfreezing its last transformer layers.

### 3.4 Encouraging Prompt Diversity with Regularization

To explicitly promote semantic diversification among the prompt embeddings, we introduce a Diversity Loss $L_{div}$. Given K embeddings from distinct prompts, we compute pairwise cosine similarities among these embeddings. Specifically, we calculate K(K-1) similarity scores, averaging them to yield a single scalar representing the mean similarity. By minimizing this similarity, we encourage each prompt embedding to capture unique semantic aspects effectively. The diversity loss is thus: $L_{div} = \frac{1}{K(K-1)} \sum_{i \neq j}^{K} CosSim(Emb_i, Emb_j)$.

### 3.5 Negation-Aware Prompt Embedding

Vision-language models often struggle with distinguishing fine-grained semantic differences or explicitly negated scenarios, as typical training does not encourage embeddings to clearly represent these contrasts. Without explicitly guiding the model to differentiate between what the image does and does not represent, embeddings may become overly generalized or ambiguous. To enhance the discriminative capabilities of vision-language embeddings, we propose a Negation Prompt Embedding strategy. Our approach explicitly generates embeddings representing semantic concepts excluded or negated by the original prompt. Specifically, we construct structured negation prompts such as:

```
[input_text]. The [APT-i] of this image does NOT mean:"
```

This explicit negation approach aims to encourage embeddings that explicitly capture semantics irrelevant to or diverging from the original input. We create negation prompts corresponding directly to each of the K original prompts, resulting in K additional negation embeddings $\{n_i\}$. Each negation embedding undergoes dimensional projection to D/K and are then concatenated, forming a single negation embedding of size D.

Specifically, for each vision embedding $q_i$, we contrast it both the original text embeddings $p_i$, and their corresponding negation embeddings $n_i$. This effectively doubles the number of text embeddings in the contrastive training. The negation-aware contrastive loss $L_{neg}$ (image-to-text only) is computed as: $L_{neg} = -\frac{1}{B} \sum_{i=1}^{B} \log \left( \frac{\exp(q_i p_i / \tau)}{\sum_{j=1}^{B} (\exp(q_i p_j / \tau) + \exp(q_i n_j / \tau))} \right)$.

Consequently, this encourages the vision-language model to differentiate between accurate and negated semantic content, improving semantic discrimination and retrieval accuracy.

### 3.6 Training Objective

Our total loss function integrates the standard CLIP contrastive loss with the diversity loss and negation-aware loss: $L_{total} = L_{con} + \alpha \cdot L_{div} + \beta \cdot L_{neg}$. We set the coefficients $\alpha$ and $\beta$ to 0.1. Our experiments show this combined objective further improves semantic alignment.

### 3.7 Video-Text Contrastive Learning

To extend our approach to video-text alignment, we adapt the vision encoder to handle video inputs while keeping the text encoder unchanged. Each video frame is independently processed by the ViT encoder, yielding token embeddings of shape $T \times N \times D$, where $T$ is the number of frames and $N$ is the number of tokens per frame. To incorporate temporal information, we add learnable temporal positional encodings of shape $T \times 1 \times 1$ before flattening the frame-wise tokens into a sequence of $TN \times D$. These are then

passed through the same attention pooling layer used in the image encoder, resulting in a single video-level embedding of dimension $D$. The text encoder continues to leverage our multi-prompt embedding mechanism, allowing it to produce rich, context-adaptive representations aligned with the video content.

# 4 Experiments

We evaluate our context-adaptive multi-prompt embedding method on image-text and video-text retrieval tasks. Our experiments include comparisons with CLIP baselines and ablation variants to demonstrate the contribution of our approach.

## 4.1 Implementation Details

We train contrastive image-text models from scratch using ViT-B/16 as the image encoder (D = 768) and LAION (Schuhmann et al., 2022) as the training dataset. Images are resized to 224×224, with visual embeddings obtained via attention pooling. The vanilla CLIP uses a 12-layer Transformer text encoder, following standard practices of Radford et al. (2021); Yu et al. (2022). In our method, we replace the text encoder with the pretrained Gemma 2B LLM (Team et al., 2024), freezing all layers except the last L Transformer layers. We use L=2 for Gemma 2B backbone. Due to the model's causal attention, we apply last-token pooling with structured prompts. The output of the LLM text encoder is first passed through a linear projection layer to reduce its feature dimension to $D/K$, where $D$ is the embedding dimension of the ViT visual encoder and $K$ is the number of adaptive prompts. We use AdamW with a learning rate of 5e-4, linear warmup for 10k steps, training for 500k iterations. We use batch size 1024 unless otherwise noted. For video-text training, we initialize from the pretrained image-text CLIP and finetune on VideoCC3M (Nagrani et al., 2022) for 50k steps with a learning rate of 1e-5, batch size 128, and 16 uniformly sampled frames per video. We evaluate under zero-shot settings on Flickr30K (Plummer et al., 2015) and MSCOCO (Chen et al., 2015) for image-text retrieval, and MSR-VTT (Jun Xu & Rui, 2016) for video-text retrieval.

## 4.2 Ablation Studies

We perform several ablation experiments to demonstrate the contribution of our context-adaptive prompting strategy. We use Gemma-2B text encoder and batch size 1024 unless otherwise noted.

**Baseline Comparisons.** As shown in Table 1, replacing the vanilla CLIP text encoder with a fully frozen pretrained LLM leads to a substantial drop in retrieval performance, likely due to insufficient adaptation to the vision-text alignment objective. Allowing partial adaptation by unfreezing the last transformer layers of the LLM helps reducing this gap. Additional gains are observed by applying prompt-based last-token pooling using the template `[input_text]. This image means: "`. Our method also builds on this prompt-based last pooling approach, extending it with multiple context-adaptive prompts.

**Number of Prompts (K).** Introducing our context-adaptive multi-prompt embedding leads to clear performance improvements. Our method uses structured prompts of the form `"[input_text]. The [APT-i] of this image means:"`, where each adaptive prompt token (`[APT-i]`) dynamically captures distinct semantic aspects. As shown in Table 2, retrieval accuracy significantly increases as the number of prompts grows 1 to 6. We select K=6, since further increases yield minimal additional benefit.

**Prompt Design and Text Embedding Construction.** Table 3 presents ablations on prompt structure and embedding construction strategies. All variants use K=6. Using a single shared `[APT]` token across all K prompts leads to redundant representations and yields limited performance similar to using a single prompt (K=1). Employing K manually crafted fixed prompts (e.g. `[input_text]. The main category of the image means:"`, or `[input_text]. The primary object in the image means:"`) improves performance, but remains suboptimal

| backbone | pooling | Flickr R@1 | | MSCOCO R@1 | |
|---|---|---|---|---|---|
| | | img-to-txt | txt-to-img | img-to-txt | txt-to-img |
| Vanilla CLIP | mean | 61.4 | 43.7 | 38.3 | 24.0 |
| LLM-frozen | last | 41.4 | 28.6 | 21.6 | 13.4 |
| LLM-unfrozen last layers | mean | 60.6 | 43.4 | 36.6 | 23.4 |
| LLM-unfrozen last layers | last | 54.0 | 37.8 | 32.6 | 20.3 |
| LLM-unfrozen last layers | prompt-last | 54.5 | 38.0 | 32.8 | 20.8 |

Table 1: **Baseline image-text retrieval performance.** We compare vanilla CLIP and CLIP variants with a pretrained LLM text encoder.

| # Prompts (K) | Flickr R@1 | | MSCOCO R@1 | |
|---|---|---|---|---|
| | img-to-txt | txt-to-img | img-to-txt | txt-to-img |
| 1 | 54.7 | 38.1 | 32.8 | 20.9 |
| 3 | 60.3 | 42.1 | 36.7 | 23.0 |
| **6** | **66.0** | **47.1** | **41.0** | **25.2** |
| 12 | 66.1 | 47.1 | 40.8 | 25.3 |

Table 2: **Effect of the number of adaptive prompts (K)**.

| method | Flickr R@1 | | MSCOCO R@1 | |
|---|---|---|---|---|
| | img-to-txt | txt-to-img | img-to-txt | txt-to-img |
| Shared [APT] token | 54.6 | 38.3 | 33.0 | 20.8 |
| Fixed prompts | 64.1 | 44.7 | 39.7 | 24.3 |
| Minimal prompts | 63.0 | 43.5 | 38.9 | 24.1 |
| Averaged K embeddings | 56.8 | 39.2 | 35.1 | 22.3 |
| Context-adaptive prompts (Ours) | **66.0** | **47.1** | **41.0** | **25.2** |

Table 3: **Ablation on prompt design and embedding construction**. All methods use K=6.

compared to our adaptive prompts. We further evaluate a minimal variant that removes contextual phrasing, using a simplified format: `[input_text]. [APT-i]:`". Although this setup performs competitively, it underperforms the full structured version, highlighting the utility of explicitly guiding the LLM to extract "[APT-i] of the image". We also study a variant where K adaptive prompt embeddings (each of size D instead of D/K) are averaged element-wise instead of concatenated. While this approach brings some improvement over a single prompt, it still underperforms our method. In contrast, our channel-wise concatenation of K adaptive prompt embeddings achieves the best performance. This design allows each [APT-i] to specialize and align with distinct channel segment of the visual embedding, encouraging semantically diverse alignment as further shown in Sec. 4.5.

**Prompt Diversity Regularization.** We study the effect of our diversity regularization loss $L_{div}$ in Table 4. Introducing moderate diversity regularization improves retrieval performance, demonstrating the benefit of explicit encouraging diversity among adaptive prompts. We set the regularization weight $\alpha = 0.1$, as higher values do not yield additional improvements.

**Negation-Aware Embedding and Loss Combination.** In Table 5, we evaluate the impact of incorporating negation-aware embeddings and the loss $L_{neg}$. Adding this negation embedding improves retrieval performance, showing that explicitly modeling semantic negation improves semantic discrimination. Combining both negation embedding and diversity regularization further boosts performance, achieving our best overall results.

**LLM Text Encoder Size.** Table 6 presents the effect of increasing the size of the LLM backbone from Gemma-2B to Gemma-9B. Using a larger pretrained language model clearly improves retrieval results.

**Contrastive Batch Size.** Table 7 studies how our method scales by varying contrastive batch sizes. our method consistently outperforms the vanilla CLIP baseline across different batch sizes (1024 and 4096). Increasing the batch size to 4096 clearly improves retrieval

| method | Flickr R@1 | | MSCOCO R@1 | |
|---|---|---|---|---|
| | img-to-txt | txt-to-img | img-to-txt | txt-to-img |
| $\alpha = 0$ (w/o $L_{div}$) | 66.0 | 47.1 | 41.0 | 25.2 |
| $\alpha = 0.1$ | **66.8** | **48.2** | **41.7** | **25.9** |
| $\alpha = 1.0$ | 65.8 | 47.3 | 41.2 | 25.6 |

Table 4: **Token diversity regularization loss ($L_{div}$).**

| method | Flickr R@1 | | MSCOCO R@1 | |
|---|---|---|---|---|
| | img-to-txt | txt-to-img | img-to-txt | txt-to-img |
| $L_{con}$ | 66.0 | 47.1 | 41.0 | 25.2 |
| $L_{con}$ & $L_{neg}$ | 67.2 | 48.0 | 41.8 | 26.0 |
| $L_{con}$ & $L_{div}$ & $L_{neg}$ | **68.3** | **48.6** | **42.3** | **26.4** |

Table 5: **Negation-aware embedding ($L_{neg}$) and loss combination.**

| LM backbone | Flickr R@1 | | MSCOCO R@1 | |
|---|---|---|---|---|
| | img-to-txt | txt-to-img | img-to-txt | txt-to-img |
| Gemma-2B | 68.3 | 48.6 | 42.3 | 26.4 |
| Gemma-9B | 70.3 | 52.7 | 44.8 | 27.6 |

Table 6: **Effect of text encoder size.**

performance, highlighting our method's capability to effectively leverage large batch size in contrastive learning.

**Number of Trainable LLM Layers.** Table 7 also examines the effect of the number of unfrozen LLM layers ($L$), with and without the learnable vocabularies (tokenizer weights). On top of unfreezing a few last layers, making the vocabulary learnable provides additional improvements. This shows that allowing more of the LLM to adapt to the contrastive objective is beneficial, though a balance must be struck with computational cost.

### 4.3 Image-Text Retrieval Results

We further evaluate the scalability of our method on image-text retrieval by increasing the contrastive batch size to 16384. For reference, OpenAI CLIP (Radford et al., 2021) leverages an even larger batch size of 32768. As shown in Table 8, our method continues to benefit from this scaling, achieving stronger performance than baseline models. We also include comparisons with other ViT-B-based CLIP methods, highlighting the effectiveness of our approach to capture diverse semantic signals for vision-language alignment.

### 4.4 Video-Text Retrieval

We extend our approach to video-text retrieval by initializing from the image-text pretrained model (with Gemma-2B as text encoder) and further training on the VideoCC3M (Nagrani et al., 2022) dataset. We evaluate on the MSR-VTT (Jun Xu & Rui, 2016) benchmark under a zero-shot setting. As shown in Table 9, our method brings substantial improvements compared to the vanilla CLIP baseline, demonstrating its effectiveness in aligning video and text through semantically diverse prompt embeddings.

### 4.5 Attention Visualization

Our text embedding is constructed by concatenating K adaptive prompt embeddings along the channel dimension. During contrastive training, the textual and visual embeddings are matched through element-wise dot product, aligning each prompt embedding with a corresponding channel segment of the visual embedding. To better understand how these adaptive prompts evolve, we visualize attention patterns from the final attention pooling

| batch size | # unfrozen layers ($L$) | Learn. vocab. | Flickr R@1 img-to-txt | txt-to-img | MSCOCO R@1 img-to-txt | txt-to-img | ZS INet Top-1 acc. |
|---|---|---|---|---|---|---|---|
| 1024 | Vanilla CLIP | Y | 61.4 | 43.7 | 38.3 | 24.0 | 53.1 |
| 1024 | 0 | N | 46.5 | 32.2 | 27.6 | 17.7 | 44.2 |
| 1024 | 2 | N | 68.3 | 48.6 | 42.3 | 26.4 | 52.4 |
| 1024 | 0 | Y | 65.1 | 46.2 | 39.1 | 24.5 | 52.3 |
| 1024 | 1 | Y | 66.6 | 47.9 | 42.5 | 26.3 | 53.5 |
| 1024 | 2 | Y | 69.1 | 49.6 | 43.5 | 27.1 | 54.0 |
| 1024 | 8 | Y | 75.4 | 53.4 | 48.3 | 29.6 | 55.9 |
| 4096 | Vanilla CLIP | Y | 81.0 | 60.2 | 51.5 | 34.8 | 67.3 |
| 4096 | 2 | Y | 81.3 | 63.9 | 54.6 | 36.1 | 67.7 |
| 4096 | 8 | Y | 84.3 | 66.2 | 60.4 | 39.4 | 68.1 |

Table 7: **Effect of contrastive batch size, number of unfrozen last layers ($L$), and learnable vocabularies.** K = 6 is used.

| method | Flickr R@1 img-to-txt | txt-to-img | MSCOCO R@1 img-to-txt | txt-to-img |
|---|---|---|---|---|
| OpenAI CLIP-B (Radford et al., 2021) | 81.9 | 62.1 | 52.4 | 33.1 |
| LongCLIP-B (Xiao et al., 2024) | 85.8 | 70.6 | 56.9 | 40.9 |
| E5-V (Jiang et al., 2024a) | 79.5 | 67.8 | 51.6 | 41.2 |
| JinaCLIP-B (Xiao et al., 2024) | 80.6 | 67.4 | 55.6 | 41.1 |
| Ours | 84.7 | 68.7 | 58.5 | 41.4 |

Table 8: **Zero-shot image-text retrieval and classification.** Batch size 16k, K=6, L=2 is used.

| method | MSR-VTT R@1 text-to-video | video-to-text |
|---|---|---|
| OpenAI CLIP-B (Radford et al., 2021) | 23.3 | 43.3 |
| SocraticModel-B (Zeng et al., 2022) | - | 46.9 |
| CLIP4Clip (Luo et al., 2022) | 32.0 | - |
| VideoCoCa-B (Yan et al., 2022) | 31.2 | - |
| Vanilla CLIP (Sec. 3.7) | 31.6 | 45.1 |
| Ours (Sec. 3.7) | 35.8 | 48.7 |

Table 9: **Zero-shot video-text retrieval**.

layer of the ViT encoder. By dividing the visual embedding channels into K segments, we obtain K attention maps, each averaged over all attention heads in its segment. We use 384×384 input images (24×24 resolution attention maps) for visualization. We observe that different adaptive prompts focus on different semantic areas: some prompts emphasize subjects (e.g. [APT-1]), others highlight relevant objects (e.g. [APT-2]), and some capture broader contextual background elements.

## 5 Conclusion

We presented Context-Adaptive Multi-Prompt Embedding, a novel approach for enhancing textual representations in vision-language contrastive learning. Our method leverages pretrained LLMs to generate multiple prompt-guided embeddings that dynamically adapt during training, allowing for richer semantic alignment with visual content. By introducing context-adaptive prompt tokens, prompt diversity regularization, and negation-aware embeddings, we improve the model's ability to capture discriminative semantics. Extensive experiments on image-text and video-text retrieval benchmarks demonstrate the effectiveness of our approach.

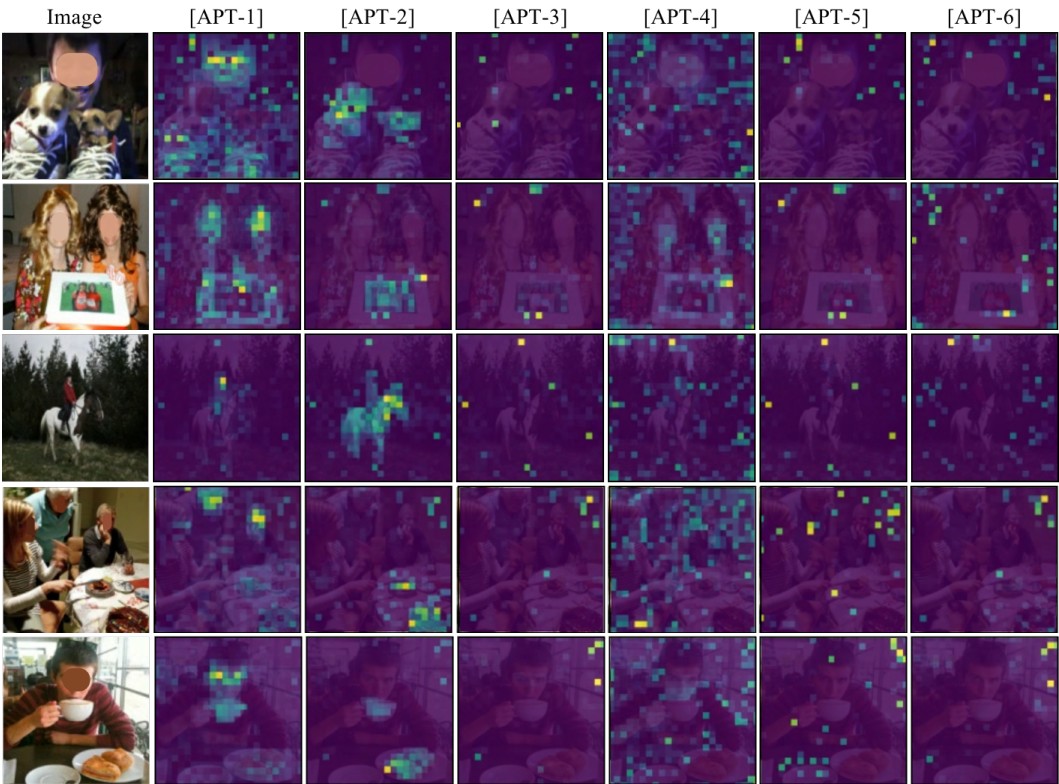

Figure 2: **Attention maps from the visual encoder corresponding to each adaptive prompt token [APT-i].** For each input image (row), we visualize the attention from the ViT attention pooling layer segmented by the K=6 prompt-specific embedding channels (columns). Each map shows how a specific APT-i attends to different spatial regions such as objects, subjects, or background, reflecting diverse visual-textual alignments.

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
