# OpenReview forum: "Context-Adaptive Multi-Prompt Embedding with Large Language Models for Vision-Language Alignment"
_colmweb.org/COLM/2025/Conference — COLM 2025_

### Official Review · Reviewer_JaD2 · 2025-05-11

**Rating:** 6
**Confidence:** 4
**Ethics Flag:** 1

**Summary:**

This paper introduces Context-Adaptive Multi-Prompt Embedding, a new method for enhancing semantic representations in vision-language contrastive learning. Departing from traditional CLIP-style models that rely on a single text embedding, the proposed approach uses multiple structured prompts, each with an adaptive token designed to capture distinct semantic aspects of the text. These prompts are processed in a single forward pass using a pretrained LLM via prompt-wise masked attention. The resulting embeddings are combined into a unified text representation, allowing for more semantically rich alignment with visual inputs. To further enhance diversity and representation quality, the method incorporates a diversity regularization loss and a negation-aware loss with negated prompts, which encourage semantic specialization among prompts and improve contrastive learning performance.

**Questions To Authors:**

Could you provide a comparison of the computational costs (e.g., memory and FLOPs) between your proposed method and CLIP-based methods?

Could you evaluate your method using other base models, such as LLaMA or similar LLMs, to demonstrate its robustness and generalizability across different architectures?

**Reasons To Accept:**

1. This paper introduces a multi-prompt framework with adaptive tokens, enabling finer-grained semantic modeling than single-prompt approaches, which leverages prompt-wise masked attention in a single forward pass, making the method efficient while maintaining expressive power.

2. The proposed method incorporates both diversity regularization and negation-aware loss, boosting semantic discrimination and contrastive learning effectiveness.

**Reasons To Reject:**

1. The proposed method introduces additional computational cost due to the forward pass through the LLM, which is significantly heavier than CLIP. This overhead should be quantified and compared to baseline models to assess its feasibility in practical settings.

2. The current evaluation is limited to a single LLM variant (Gemma). To demonstrate generalizability and robustness, the method should be tested with a wider range of base models, such as LLaMA, Qwen, and other widely used LLMs.

---

> ### Author Response · Authors · 2025-06-03
> **Rebuttal by Authors**
>
> We thank Reviewer JaD2 for the review and constructive suggestions. Our response is detailed below.
>
> > ***1. Computational cost***
>
> Our method fine-tunes only the final layer of the Gemma-2B model. For training efficiency, we cache hidden states from the frozen LLM layers (L168-169). This dramatically reduces the feed-forward FLOPs from 144.2 to 24.8, which is only marginally higher than vanilla CLIP's 23.5 GFLOPs.
> To further address feasibility and our strategy's benefits, we experimented with a 128M-parameter LM from MaMMUT (Kuo et al. 2023). Our approach again yielded clear improvements, with comparable GFLOPs (24.4 without caching) to vanilla CLIP.
>
> | method               | GFLOPs |
> | :------------------- | :----: |
> | Vanilla CLIP         |  23.5  |
> | Ours without caching | 144.2  |
> | Ours with caching    |  24.8  |
>
>
> | method (bs=1024)    | GFLOPs | Flickr (I2T) | Flickr (T2I) | COCO (I2T) | COCO (T2I) |
> | :-------------------------- | :----: | :--------------: | :--------------: | :------------: | :------------: |
> | Vanilla CLIP                |  23.5  |       61.4       |       43.7       |      38.3      |      24.0      |
> | Ours (with 128M text model) |  24.4  |       64.2       |       45.5       |      40.1      |      24.8      |
>
>
>
> > ***2. Generalizability to other LLM***
>
> We evaluated our approach using the LLaMA3 8B model as the text encoder (batch size 4096, unfrozen last layer). As shown below, our method showed consistent improvement over a single prompt baseline with the same LLaMA backbone.
>
> | Method (with LLaMA-8B, bs=4096) | Flickr (I2T) | Flickr (T2I) | COCO (I2T) | COCO (T2I) |
> | :-------------------------------------- | :--------------: | :--------------: | :------------: | :------------: |
> | Baseline (LLaMA-CLIP)                   |       79.2       |       58.5       |      50.6      |      33.7      |
> | Ours (LLaMA-CLIP with our multi-prompt) |       84.7       |       65.2       |      57.9      |      38.2      |

---

> > ### Comment · Reviewer_JaD2 · 2025-06-09
> >
> > The rebuttal addressed my concerns. Hence, I keep my original score.

---

### Official Review · Reviewer_L9Jh · 2025-05-12

**Rating:** 7
**Confidence:** 4
**Ethics Flag:** 1

**Summary:**

In the context of image-text retrieval, this paper proposes a text embedding method using multiple prompts containing special tokens that are combined to produce a text embedding (e.g., "the [SPECIAL TOKEN] of this prompt means:"). In contrast to prior methods that use static prompts, this approach allows a significant part of each prompt to be learned. The approach is trained using contrastive learning that includes negated embeddings ("... does NOT mean:") and combined with a diversity loss based on the pairwise cosine similarity between embeddings. Results on common image-caption retrieval benchmarks indicate that the proposed approach outperforms comparable CLIP baselines, despite being trained with a much smaller batch size.

**Questions To Authors:**

- How does the performance of prompt-wise attention masking compare to batching when using prefix caching? Is there some factor that should make the masking more efficient?

**Reasons To Accept:**

This is an interesting approach that addresses a shortcoming in prior work (i.e., prompts are static). Experiments demonstrate that this approach is effective for image-caption retrieval. The paper is generally easy to follow and likely could be reproduced without too much effort, though this would be easier if code and weights were made available.

**Reasons To Reject:**

While the proposed approach does outperform CLIP, the results are not particularly strong compared to SigLIP (or SigLIP2) and it is not clear why the approach was applied to image-text retrieval specifically, given that the prior work used as motivation considered text only. While this is a challenge setting, the motivation of moving from static to dynamic prompts would also apply to text.

---

> ### Author Response · Authors · 2025-06-03
> **Rebuttal by Authors**
>
> We appreciate Reviewer L9Jh for their feedback. Our response is detailed below.
>
> > ***1. Comparison with SigLIP/SigLIP2***
>
> While SigLIP and the very recent SigLIP 2 show strong performance, SigLIP's core method (replacing softmax with a sigmoid contrastive loss) and SigLIP 2's additional techniques (e.g., decoder-based pretraining, self-supervised losses, data curation) are largely orthogonal and complementary to our main contribution: a multi-prompt embedding strategy.
>
> Direct comparisons are challenging due to significant differences in training regimes. SigLIP models leverage the much larger, proprietary 10B-scale WebLI dataset (which is not publicly available), larger batch sizes (32k) and longer training (1,250k iterations), compared to our LAION experiments in Table 3 (16k batch size, 500k iterations).
> To isolate our strategy's benefits in a SigLIP-aligned setting, we conducted new experiments using a sigmoid loss with a 4096 batch size. Our multi-prompt method shows consistent gains in this sigmoid loss setting as well, indicating its benefits are independent of the specific contrastive loss function (softmax vs. sigmoid).
>
> | method                                  | Flickr (I2T) | Flickr (T2I) | COCO (I2T) | COCO (T2I) |
> | :-------------------------------------- | :--------------: | :--------------: | :------------: | :------------: |
> | Vanilla CLIP (with SigLIP sigmoid loss) |       80.6       |       60.0       |      51.7      |      34.8      |
> | Ours (with SigLIP sigmoid loss)         |       83.2       |       64.2       |      56.2      |      36.6      |
>
>
> > ***2. Motivation for image-text setting  and text-only application***
>
> To explore the reviewer's point that our approach could benefit text-only settings, we ran a preliminary experiment on the Semantic Textual Similarity benchmark (STS-B) with finetuning. This initial experiment shows encouraging results for our multi-prompt method over the baseline.
>
> | method               | STS-B (Spearman) |
> | :------------------- | :--------------: |
> | Baseline Gemma-CLIP  |       87.2       |
> | Ours Gemma-CLIP      |       88.9       |
>
>
> Our primary motivation for the vision-language (VL) framework lies in its advantages for our adaptive prompt tokens. VL contrastive learning allows these tokens to be specialized by optimizing for alignment with visual content, enabling them to capture semantic aspects highly relevant for visual matching.
> This specialization is demonstrated by our attention visualizations in Figure 2, where different [APT-i] tokens guide the visual encoder to focus on distinct image regions.
>
>
> > ***3. Prompt-wise attention masking vs. Batching with prefix caching***
>
> Both our prompt-wise attention masking and prefix caching with batching efficiently avoid recomputing KV states of the shared [input\_text] prefix. While the efficiency of batched prefix caching can depend on system-level optimizations for managing and loading KV caches for batched items, the core computational work is likely similar.
> Our prompt-wise masking provides a direct implementation, processing K prompts in a single forward pass, and is not reliant on specific device support (e.g. those with limited memory for caching) or framework optimizations for cache handling.

---

> > ### Comment · Reviewer_L9Jh · 2025-06-05
> >
> > Thanks for your response. The results and clarifications have addressed all my concerns, and I've increased my review score to reflect this.

---

### Official Review · Reviewer_TY9F · 2025-05-13

**Rating:** 6
**Confidence:** 4
**Ethics Flag:** 1

**Summary:**

This paper introduces "Context-Adaptive Multi-Prompt Embedding" to align visual data with textual descriptions. Unlike existing methods like CLIP, the method introduces multiple structured prompts, where a distinct adaptive token captures diverse semantic aspects of the input text. A pretrained LLM is used to process all the prompts all at a single pass. With the additional regularization loss for diversity and a negation-aware loss, the proposed method achieves consistent improvements on image-text and video-text retrieval tasks.

**Questions To Authors:**

Please check the concerns above. The reviewer is open to increase score if the concerns can be addressed.

**Reasons To Accept:**

1. The idea of using LLM to process both positive and negative prompts is interesting.
2. The paper is overall well presented and easy to follow.

**Reasons To Reject:**

1. The major of concern of the reviewer is on lack of proper baseline comparisons. There has been highly relevant work [a,b] on either formulating the decomposition of a prompt into fine-grained embeddings like [a] or leverage LLM to improve vision-language model [b] that needs to be more discussed and compared. For example, baselines like multiple fine-grained embeddings could be constructed in [a]'s manner through text prompts; baselines like using LLM to possibly generate more positive/negative descriptions can be provided to validate that current approach is indeed an ideal way of using LLM in vision-language representation learning.

2. Currently vanilla CLIP has a much smaller text encoder compared to the proposed one. What if use the same Gemma model as initialization and perform PEFT for it in CLIP's setup?

3. Could the proposed design bring additional different usage in inference time? This could have been explored more.

Post-rebuttal:
The reviewer increases the score as the new baselines comparisons addressed the concerns.


[a] Learning to Decompose Visual Features with Latent Textual Prompts

[b] Enhancing CLIP with GPT-4: Harnessing Visual Descriptions as Prompts

---

> ### Author Response · Authors · 2025-06-03
> **Rebuttal by Authors**
>
> We thank Reviewer TY9F for the constructive feedback and suggestions.
>
> > ***1. Comparison with [a] DeFo and [b] VDT-Adapter***
>
> We will include the discussion on [a] and [b] in the final version. While DeFo [a] and VDT-Adapter [b] use decomposed representations or LLMs, their methods, task formulation, and applicability fundamentally differ from our approach to general image-text retrieval.
>
> DeFo [a] is designed for fixed-category classification and requires fine-tuning. It learns latent textual embeddings and a classification-specific linear layer, rather than handling natural language inputs. This constrains DeFo to predefined classes and makes it unsuitable for zero-shot retrieval.
> Inspired by DeFo's M-token-per-query concept (M=16 in DeFo) and the reviewer's suggestion, we tested a baseline appending 16 [APT] tokens to the [input\_text] (i.e., "[input\_text] [APT-1]...[APT-16]"). Our structured prompt method (single [APT-i] per prompt in the template "the [APT-i] of the image means:") performs better.
>
> | method (bs=1024, K=6)        | Flickr (I2T) | Flickr (T2I) | COCO (I2T) | COCO (T2I) |
> | :----------------------------------- | :--------------: | :--------------: | :------------: | :------------: |
> | DeFo [a]-inspired (M-token-per-query) |       63.4       |       43.8       |      39.1      |      24.1      |
> | Ours                                 |       66.0       |       47.1       |      41.0      |      25.2      |
>
> VDT-Adapter [b] uses a large external LLM (GPT-4) with manual, dataset-specific heuristic prompts (e.g. for FGVC Aircraft dataset, using attributes like "manufacturer" or "engine count") to augment class names for classification, and is not evaluated on retrieval. Applying this to retrieval by augmenting every query with GPT-4 (trillion-parameter scale) would incur significant cost. More importantly, it is unclear how to design general-purpose prompts for retrieval.
> Although re-captioning the entire LAION pretraining dataset is beyond the scope of this rebuttal and is an orthogonal research area, we augmented Flickr/MSCOCO test queries using an external LLM with a prompt ("Describe the visual aspect in detail"). This did not help the zero-shot retrieval.
>
> | method (bs=1024)                     | Flickr (I2T) | Flickr (T2I) | COCO (I2T) | COCO (T2I) |
> | :------------------------------------------- | :--------------: | :--------------: | :------------: | :------------: |
> | Vanilla CLIP                                 |       61.4       |       43.7       |      38.3      |      24.0      |
> | Vanilla CLIP with LLM-augmented input text |       60.2       |       43.3       |      38.1      |      23.7      |
> | Ours                                         |       66.0       |       47.1       |      41.0      |      25.2      |
>
> > ***2. Comparison with CLIP with Gemma text encoder***
>
> To isolate our multi-prompt strategy's benefits from using a larger LLM (Gemma-2B), we note that our Table 1 already evaluates a 'LLM-unfrozen last layer' setup. This uses Gemma-2B by fine-tuning only its final layer (PEFT) and includes baselines with basic last-token and simple prompt-based last-token pooling. We further experimented with standard mean pooling for this Gemma encoder. The results show our method's gains stem from our multi-prompt approach, not just the larger LLM:
>
> | method (bs=1024)                               | Flickr (I2T) | Flickr (T2I) | COCO (I2T) | COCO (T2I) |
> | :----------------------------------------------------- | :--------------: | :--------------: | :------------: | :------------: |
> | Vanilla CLIP (mean pooling)                            |       61.4       |       43.7       |      38.3      |      24.0      |
> | Gemma-CLIP (mean pooling, unfrozen last layer)         |       60.6       |       43.4       |      36.6      |      23.4      |
> | Gemma-CLIP (mean pooling, LoRA)                        |       63.1       |       44.6       |      39.2      |      24.3      |
> | Ours: Gemma-CLIP w/ our multi-prompt (unfrozen last layer) |       66.0       |       47.1       |      41.0      |      25.2      |
>
> > ***3. Additional inference-time usage***
>
> Interpretability: Our K individual prompt embeddings and their visual attention patterns (Fig. 2) provide granular understanding of text-visual alignment.
>
> Aspect re-weighted inference:
> The K distinct [APT] embeddings allow inference-time adjustment. The similarity score $S = \sum_{k=1}^{K} (p_k \cdot q_k)$ (where $p_k$ and $q_k$ are k-th segments of normalized text/image embeddings), can be modulated by weights $w_k$ to $S_{weighted} = \sum_{k=1}^{K} w_k (p_k \cdot q_k)$.
> Inspired by Figure 2 where different [APT-i] tokens show varied focuses (salient objects, secondary elements, or background), we tested this on ImageNet zero-shot classification (batch size 4096). Up-weighting the first segment ([APT-1], presumed subject focus) with $w_1$=1.25 and slightly reducing others improved the top-1 accuracy from 61.4% to 61.8%.

---

> ### Author Response · Authors · 2025-06-10
>
> Dear Reviewer TY9F,
> We are checking in to see if our rebuttal addresses your questions. We are happy to address any others you may have. Thank you.

---

### Decision · Program_Chairs · 2025-07-08

**Decision:**

Accept

**Comment:**

The core idea of the paper is to move beyond single-text embeddings (like those in standard CLIP models) by using multiple structured prompts, each with a distinct adaptive token. These tokens are designed to capture diverse semantic aspects of the input text. The inclusion of both diversity regularization loss and negation-aware loss effectively encourages specialization across prompts and improves contrastive discrimination, leading to richer and more robust representations.

 The authors provided thorough rebuttals that addressed the initial concerns regarding baseline comparisons, computational cost, and generalizability to different LLMs. These updates should be included in the revised paper.